# Boxing Punch Detection with Single Static Camera

**DOI:** 10.3390/e26080617

**Published:** 2024-07-23

**Authors:** Piotr Stefański, Jan Kozak, Tomasz Jach

**Affiliations:** Department of Machine Learning, University of Economics in Katowice, 1 Maja 50, 40-287 Katowice, Poland; jan.kozak@ue.katowice.pl (J.K.); tomasz.jach@ue.katowice.pl (T.J.)

**Keywords:** combat sports analysis, punch detection, boxer detection, background subtraction

## Abstract

Computer vision in sports analytics is gaining in popularity. Monitoring players’ performance using cameras is more flexible and does not interfere with player equipment compared to systems using sensors. This provides a wide set of opportunities for computer vision systems that help coaches, reporters, and audiences. This paper provides an introduction to the problem of measuring boxers’ performance, with a comprehensive survey of approaches in current science. The main goal of the paper is to provide a system to automatically detect punches in Olympic boxing using a single static camera. The authors use Euclidean distance to measure the distance between boxers and convolutional neural networks to classify footage frames. In order to improve classification performance, we provide and test three approaches to manipulating the images prior to fitting the classifier. The proposed solution achieves 95% balanced accuracy, 49% F1 score for frames with punches, and 97% for frames without punches. Finally, we present a working system for analyses of a boxing scene that marks boxers and labelled frames with detected clashes and punches.

## 1. Introduction

Currently, around the globe, multiple devices are installed that allow for the recording of images and sounds. The vast majority of these are surveillance cameras but, simultaneously, more and more footage is produced by bystanders. When there is an enormous amount of material, it is nearly impossible to understand a recorded scene with no automatic solutions. Extracting valuable information is becoming an important factor in making sense of such data [1,2,3]. Currently, cameras in public spaces (e.g., airports, banks, train stations, and hospitals) analyse the behaviour of individuals in order to detect dangerous situations and issue an alert to the relevant authorities [4,5]. In private spaces, cameras automatically monitor employee behaviour and equipment to improve health and security [6,7,8]. Additionally, in industry, cameras monitor the quality of manufactured products [9,10].

Recently, the use of machine learning methods is becoming more popular in the domain of combat sports [11,12,13]. The current state-of-the-art approach is to use invasive (e.g., wearable sensors) or non-invasive (e.g., cameras) approaches to detect players and analyse their movement [14,15,16,17,18,19,20,21].

The aim of this paper is to provide an automatic way of detecting punches in Olympic boxing using computer vision algorithms. The rest of the paper will discuss the other requirements and limitations, as well as the complete preliminary solution. We are using non-invasive methods and providing a solution to a problem with close co-operation with stakeholders, who validate the approach on a daily basis. The research goal of this paper is to verify the influence of multiple different approaches to image manipulation on the performance of a punch classifier.

The remainder of this article is organised as follows. Section 2 gives a broad description of the current state of the art. Section 3 presents the proposed approaches of the authors and the process of data collection and labelling. Section 4 provides the results of the experiments that describe the efficiency of three different approaches to detecting punches. Section 5 gives a statistical analysis of the results obtained. Section 6 discusses the concept of entropy in boxing punch detection, providing insights into the uncertainty and variability in the classification process. Section 7 contains the conclusions of the paper, along with ideas for future work.

## 2. Related Works

Detecting people in images and analysing their behaviour in videos is the current focus of many computer vision studies. In many cases, these systems are used to monitor and protect our lives [6,7,8,22]. There are many monitoring systems for potential dangerous event recognition (e.g., stroke or collapse) to protect older people using RGB and RGB-D cameras [4,5]. Additionally, cameras may be used to protect passengers at airports by automatically detecting unattended bags, which may be a direct source of danger [23].

Therefore, many systems are trying to analyse a scene in order to detect and classify people’s actions. This is a complex problem because of the many possible combinations of body postures, body parameters (such as height, weight, etc.), clothing, and environment conditions where footage is captured. Many papers [24,25,26] have been dedicated to detecting pedestrians in traffic scenes, which is an important path in the development of self-driving cars.

The recognition of human actions plays a crucial role in the analysis of athletes and players in many sports [19]. Fully automatic recognition is still an open problem; refs. [27,28,29,30] describe different approaches but are still able to classify only a tiny fraction of interesting behaviour. A similar research problem can be found in object detection, where research has focused on detecting objects that represent a particular class of objects [31,32]. The current state of these systems is far from a universal solution with the possibility of understanding the context of most scenes in real life.

Moreover, cameras are essential in sports. Because of them, all live broadcasts can be transmitted to a wider audience. Furthermore, current sophisticated computer vision systems track players, analyse their movements, and generate reports for reporters, coaches, and others for further analysis [33]. Tracking systems are used to automatically focus attention on interesting areas (e.g., a place on a pitch where the action is happening). In their current state, these systems can also be used to create databases for teams that describe their movement and behaviour on the pitch. This is the premise of the extensive analysis concluded by finding the strengths and weaknesses of both the team and each individual player [34]. The knowledge collected by vision computing is then applied using a statistical and machine learning approach.

However, cameras in many sports events play a crucial role. The current systems use several cameras in stadiums to analyse players’ movements. Camera data are used to analyse players in football [35], tennis [36], and other popular sports [37,38,39]. Some systems are very complex and use up to 10 cameras that capture 340 frames per second to generate 3D models of ball trajectories in cricket games [33].

Applying similar techniques to combat sports is not as sophisticated as in the previous examples. The main reason for this is the lack of publicly available labelled data, as in other sports [40,41,42]. This is due to boxing being much less popular than football. However, current research dedicated to the analysis of boxing fights is using one of two approaches. One of them [14,15,16,17] is considered an “invasive” method, which uses wearable sensors to monitor boxer behaviour with special consideration for detecting punches. The main advantage of this approach is the accuracy and purity of the collected data, but it may be dangerous to boxers and will be banned by game regulations, as happened in NBA games [33]. The second way, which is a “non-invasive” method, uses RGB or RBG-D cameras to analyse boxers’ movement, detect events in the boxing ring, and classify punches [18,19,20,21]. In addition, all the relevant works on boxing fight analysis and the years of their publication are presented in Figure 1.

### Motivation for the Study

The motivation for our study arises from the need to improve the analysis of boxing performance using non-invasive methods. Unlike more popular sports, such as football, boxing lacks publicly available labelled data, which makes it difficult to grow robust analytical systems [40,41,42]. Traditional methods for monitoring boxer performance often rely on wearable sensors, which, despite their accuracy, pose safety risks and are sometimes prohibited in competitions [14,15,16,17]. Non-invasive methods using RGB or RGB-D cameras offer a safer alternative but are currently underdeveloped for real-time application in boxing [18,19,20,21].

Our proposed system aims to fill this gap by providing a real-time, non-invasive solution for punch detection in Olympic boxing using a single static camera. This approach is designed to be cost-effective and accessible, reducing the need for complex and expensive multi-camera setups. By leveraging advanced image manipulation techniques and convolutional neural networks, our method enhances the accuracy of punch detection while ensuring the safety and comfort of athletes.

## 3. Methodology

The main goal of this paper is to propose a system to detect punches in Olympic boxing. The current state of the system uses a single camera in front of the boxing ring. As will be discussed in more detail later, some of the limitations of the current approach might be overcome by using multiple cameras. The system combines the following two modules:Clash detecting—An approach based on boxer detection [43] and a measurement of the distance [44] between them using Euclidean distance to detect possible situations with punches.Punch detecting—An approach that uses neural networks to classify frames and detect the actual moment when punches hit.

The main purpose of the detection of clashes is to reduce the amount of data before the data labelling process (which was explained in Section 3.2). Reducing data also has a positive impact on processing new videos; it might be considered as a filter that reduces moments where boxers were standing far apart without any chance of punches. This reduces the processing time and computing power needed to actually detect punches, as well as decreases the number of false positives. The exact method of clash detection was previously described in the authors’ paper [44].
(1)$$d\left(A,B\right)=\sqrt{{\left(x_{B}-x_{A}\right)}^{2}+{\left(y_{B}-y_{A}\right)}^{2}},$$
where *A* and *B* are points in the co-ordinate system; *x* and *y* stand for co-ordinates for values of the points.

In the second step of the presented approach, only frames with boxers in close proximity to each other are used. Punch detection is a binary classification with two classes (punch and not punch). In order to classify this, an approach using neural networks was used. The authors used the CNN neural network, which was inspired by natural biological visual recognition mechanisms [45] and is popular in the domain of image classification. Furthermore, the authors suspect that convolutions allow one to emphasise small details which are vital for clash detection. In order to train classifiers, the novel neural network structure was created with fewer layers compared to popular network structures such as ResNet, Inception, etc. The main reason is to speed up the training process and to overcome the limitations with computing power. The structure of the convolutional neural network used is presented in Figure 2.

The input frames were resized to 180 × 180 pixels to balance computational efficiency and model performance. While the original video frames were recorded in full HD resolution, processing such high-resolution images would significantly increase the computational load and training time, especially given the extensive dataset. Resizing to 180 × 180 pixels ensures that the model can be trained more rapidly without requiring prohibitively high computing power. This dimension was selected as it is sufficiently large to capture the essential features necessary for punch detection while being small enough to allow for faster processing and iteration during model training. Additionally, this resizing step helps standardise the input data, which is crucial for the stability and performance of the convolutional neural network used for classification. By reducing the resolution, we ensure that the model focuses on the most relevant parts of the image, thereby maintaining the ability to emphasise small but vital details for accurate punch detection.

Punch detection is one of the computer vision problems in which algorithms must learn how to detect very small objects in an image. It is our assumption that classification without preprocessing will not work properly. The basis of this was covered by previous works [1,2,3,46], where adapting the input image resulted in better classification results. The confirmation of this hypothesis applied to punch detection will be verified by the performance of classification. We propose different methods for manipulating the images prior to fitting the classifier to check the performance of the classification over the same dataset but preprocessed using different methods.

The authors proposed and tested four approaches, three of which were created in order to extract ROIs (regions of interest). The following approaches are the subject of this research.

Original image (Approach 1)—The algorithm receives the original image without any transformations.Colour extraction (Approach 2)—Based on the representation of the HSV [47] image model, we extract the blue and red colours of the boxers’ outfits. The outfit colours are regulated by the rules of Olympic boxing and are constant in every fight. According to these rules, a boxer must wear red or blue gloves, a head guard, and a singlet corresponding to their respective corner.Background subtraction (Approach 3)—The algorithm, which is based on the work in [48], removes static objects and elements from the video, leaving only objects in motion.Hybrid method (Approach 4)—Combining in sequence the previous two approaches: removing colours other than blue or red followed by the removal of static elements such as the floor or corners of the boxing ring.

Example results of video frame segmentation using all four proposed approaches are presented in Figure 3. In addition, we provided a flowchart of the entire processing pipeline proposed in this study in Figure 4. The pipeline starts with the data recording step (which is described in Section 3.1) and ends with the evaluation classification performance step (which is described in Section 4).

### 3.1. Data Collection

For sports image classification problems, it is difficult to find good quality and valuable datasets. This is even more difficult for boxing fight footage, especially for videos verified and labelled by experts (e.g., boxing referees with a licence). Therefore, we need to record, collect, and label the relevant sets of data.

The necessary footage of the boxing fights was recorded in Poland in the Silesian league for juniors, cadets, and seniors. For this purpose, four GoPro Hero8 cameras with power banks and 128 GB memory cards were used. Due to the dynamic nature of fights and the occlusions of clashes, each piece of camera footage can provide a unique opportunity to detect punches. The cameras were mounted behind each corner on 1.8 m high tripods and recorded video in full HD resolution at 50 frames per second. After the competition, which lasted four hours, each memory card was nearly full, totalling just under 500 GB of recorded footage [44].

The proposed recording setup is presented in Figure 5 and is inspired by the positions of three referees behind the boxing ring. Four cameras were used in order to avoid occlusions between boxers and referees in the boxing ring. The combined footage allows for observing the fight and capturing more details than the three referees can capture on their own. In this paper, the authors have only used footage from one camera but, in the future, it will be combined and used for ensemble voting in the classification process.

### 3.2. Data Labelling

The labelling process was difficult due to the huge amount of data. In order to prepare the videos for image classification, one has to treat each frame individually. Because a punch happens rather quickly, precision in labelling is crucial. Four hours of recording with four cameras in 50 frames per second gives about three million frames to label; thus, reducing the amount of data to a more practical volume is imperative. Clash detection provides good results and filters out about 70% of the footage where the boxers were standing far apart without any chance of punches.

The filtered footage was loaded into a labelling tool (CVAT—https://github.com/opencv/cvat (accessed on 15 May 2024)) and passed through the labelling process commenced by a licensed boxing referee. The referee provided information on images from two cameras. The whole fight was divided into 14 videos for each camera, totalling 28 video clips. Each video contains about 12 min (before the filtering process) of recording. As a result, the referee looked at 312,774 frames, where 11,345 (roughly 3.62%) were considered as punches and 301,429 were not. This shows that the number of frames with a punch is significantly disproportional to the number of frames without a punch. An example of a labelled video frame is presented in Figure 6, where the referee labelled the frame as the “punch” class and also drew a bounding box around the punch area.

In order to address reproducibility, we made our dataset publicly available. The dataset, which includes the labelled boxing footage used in this study, is accessible on Kaggle (https://www.kaggle.com/datasets/piotrstefaskiue/olympic-boxing-punch-classification-video-dataset (accessed on 15 May 2024)).

### 3.3. Public Release of Data

The collected and prepared data were made available for noncommercial use: in academic institutions for teaching and research purposes and in non-profit research organisations. Along with the release, a detailed description related to the data was prepared, such as recording methodology, labelling methodology, and privacy and data protection [49].

The entire competition (the inaugural boxing league of youngsters, cadets, and juniors organised in 2021 in Szczyrk) lasted approximately four hours and the recorded material took up almost 500 GB of disk space. As interpreted by the legal department of the University of Economics in Katowice, in order to ethically and legally make the constructed database public, care was taken to reliably blur the faces of those appearing in the footage to protect personal data, using available algorithms.

## 4. Experiments

The results of the experiments were obtained on a computer using an Intel Core i9-11900K@3.50 GHz 16-core processor, 64 GB of RAM (from Polish company GOODRAM, Łaziska Górne, Poland), and an Nvidia Geforce GTX 1080Ti graphic card, using the Ubuntu 22.04.4 operating system.

In order to test our hypothesis, a binary classification machine learning model using a neural network was created. In order to confirm the statistical performance of the classification, each approach was trained and evaluated 30 times using a random sub-sample from the whole dataset. Therefore, in Section 4, the authors presented medians of the computed performance metrics: computed accuracy (Equation (Equation 2)), balanced accuracy (Equation (Equation 5)), precision (Equation (Equation 4)), and recall (Equation (Equation 3)). As the number of video frames with punches is in the order of several magnitudes lower than the frames without punches (the data is highly imbalanced), balanced accuracy should give a better perspective for performance evaluation.
(2)$$Accuracy=\frac{TP+TN}{TP+TN+FP+FN},$$
(3)$$Recall=\frac{TP}{TP+FN},$$
(4)$$Precision=\frac{TP}{TP+FP},$$
(5)$$BalancedAccuracy=\frac{Recall+TNR}{2},$$
where

TP is the true positive, which denotes the number of frames correctly classified as a punch;TN is the true negative, which denotes the number of frames correctly classified as no punch;FP is the false positive, which denotes the incorrectly classified frames as a punch;FN is the false negative, which denotes the incorrectly classified frames as no punch;TNR is the true negative ratio, defined as $\frac{TN}{TN+FP}$, also known as specificity or selectivity.

In total, we trained four classifiers using transformed datasets according to our methodology (Section 3). Each model in the input layer obtains images at a 180 × 180 resolution with a colour channel; in addition, the classifiers were trained with an Adam optimiser and cross-entropy loss function. The entire dataset was divided into two subsets: 80% of the samples were used for training and the remaining 20% were used for testing purposes. The model for each approach was trained and evaluated 30 times using a random subsample from the whole dataset to statistically confirm the classification performance. Therefore, Table 1 and Table 2 contain the medians of the computed metrics.

As expected, Table 1 and Table 2 and Figure 7 show that the approach to detect punches in the original image without any preprocessing steps is not optimal. The model is very unstable, and the classifier repeatedly produces a constant, single-class output. This is also confirmed by the recall results presented in Figure 7 and the FN metric in Table 2 for this approach. The problem of the classifier bias to output a single class is the main challenge in working with imbalanced data. In the case of our problem, this is, therefore, convergent with most real-world computer vision problems. This is why we also use balanced accuracy in the analysis and maintained diligence in minimising imbalance during data preparation.

The performance metrics for the classifier using the unprocessed images are presented in Figure 7. We treat them as the reference point for other performance metrics presented separately in Figure 8, Figure 9, Figure 10, Figure 11, Figure 12, Figure 13, Figure 14 and Figure 15. Additionally, to improve readability, the outliers were filtered using the three-sigma rule [50] shown in Equation (Equation 6). The number of removed observations was noted in each figure caption.
(6){a−3σ<X<a+3σ}
where *a* is the arithmetic mean and σ is the standard deviation.

In addition, each proposed method of segmentation was tested for processing performance on a subset that contained 14,331 frames. The results are presented in Table 3. The approach based on colour extraction (Apr. 2) proved to be the fastest and most computationally efficient. The hybrid approach (Apr. 4), which combines the two following approaches—colour extraction (Apr. 2) followed by the removal of static elements (Apr. 3)—proved to be the slowest, as expected.

Figure 16 and Figure 17 present a complete visualisation of the individual steps during punch detection. The former shows the close-combat situation with no punches, while the latter clearly informs about the hit. Each subframe on these figures corresponds to a single step in the proposed system, from the original image through the detection mask (obtained based on Approach 3, as described in Section 3), up to the final visualisation with labels applied directly to the frame. The label “NEAR” printed in yellow means that the boxers are close together, while the label “PUNCHES” denotes the detected punches between the boxers in the current scene.

Figure 17 and Figure 18 show the state of the fight when there is no contact between the boxers. In one of them, the boxers are only close together without any active fight with punches; therefore, only the “NEAR” label was printed. In the second situation, the boxers move to their own corners at the end of the round and stand far apart; therefore, no label is written on them.

The results (Table 1) have shown that Approach 3 (background subtraction) achieved the best scores using half of the metrics. Approach 3 achieved nearly the same balanced accuracy and precision for “no punch”, whereas Approach 1 provided slightly better recall for “punch” and F1 score for “not punch”. What is surprising is that the best medians of the results in these categories are achieved by the classifier using unprocessed images. That leads to the conclusion that further research on image processing might result in even better images. At the same time, the numerical stability of the unprocessed image classifier is far from optimal and, thus, not suitable for general use.

Therefore, it can be considered that Approach 3 has the best impact on classification performance. This is quite surprising because Approach 4 was the most sophisticated. In order to remove the referee from the scene, which is proven to work (Figure 3), extra processing steps are taken, which should make the scores better. Based on the results from Table 1, Approach 4 (hybrid method) is slightly worse and sometimes as good as the best method: Approach 3. It is good to note that Approach 4 is an extension of Approach 2, which effectively improves the metrics score. Therefore, one might come to the conclusion that colour extraction does remove too much valuable information from the scene.

## 5. Statistical Analysis

The experimental results of the proposed approaches were compared using a non-parametric statistical hypothesis test, i.e., the Friedman test [51,52] for α=0.05. The parameters of the Friedman test are presented in Table 4. The same table shows the mean rank values for the compared approaches. The results for each of the classification quality measures analysed were used for statistical tests.

The highest rank (1.625) was obtained for Approach 3 and it is, at the same time, critically better than Approach 1 (rank difference of 1.6250, with a 5% critical difference of 1.2274). At the same time, this is the only critical difference between all approaches. Because no approach was critically worse than all the other approaches, we did not perform a second round of statistical analysis.

Very similar ranks were obtained for Approach 2 and Approach 4: 2.5000 and 2.6250, respectively. These are lower ranks than Approach 3 (by 0.875 and 1.000), but there is no critical difference between the approaches in this case.

In summary, as a result of the statistical analysis, Approach 3 turns out to be statistically the best, is better (but not critically better) than Approaches 2 and 4, and is critically better than Approach 1. In contrast, Approaches 2 and 4 are better than Approach 1.

## 6. Entropy in Boxing Punch Detection

Entropy, a concept rooted in thermodynamics and information theory, measures the degree of disorder or uncertainty in a system. In the context of data analysis, entropy quantifies the amount of unpredictability or randomness in a dataset. For a machine learning model, particularly in classification tasks, understanding and managing entropy can provide valuable insights into ensuring accurate and reliable predictions.

When analysing video frames for punch detection in boxing, each frame can be considered a discrete data point. The goal is to classify each frame accurately as either containing a punch or not. In this context, entropy can be useful for evaluating the uncertainty in the classification process and improving the model’s robustness.

Entropy *H* can be calculated using formula (Equation 7).
(7)$$H=-\sum_{i=1}^{n}p_{i}\log_{2}p_{i},$$
where $p_{i}$ is the probability of occurrence of class *i* in the dataset. In our case, we have two classes: ”punch” and ”no punch”. The probabilities are derived from the frequency of these classes in the training data.

Given the initial data
$p_{\mathrm{punch}}=\frac{11,345}{312,774}\approx 0.0362$$p_{\mathrm{not}\;\mathrm{punch}}=\frac{301,429}{312,774}\approx 0.9638$
the initial entropy can be calculated as
$$H=-(0.0362\log_{2}0.0362+0.9638\log_{2}0.9638)\approx 0.2244.$$

This low entropy indicates a high imbalance in the data, reflecting low uncertainty in predicting the dominant class (“not punch”) but potentially high misclassification rates for the minority class (“punch”).

Monitoring entropy during the training phase can help in understanding how well the model is learning the distinguishing features of punches. A decrease in entropy over successive epochs indicates that the model is becoming more confident in its predictions. For instance, if the model starts with an entropy of 0.2244 (initial low uncertainty due to imbalance) and reduces to 0.1 (even lower uncertainty), it signifies that the model is learning effectively. However, if the entropy remains high, it suggests that the model is struggling to learn, possibly due to insufficient or poor-quality data.

Incorporating entropy into the analysis of boxing punch detection with a single static camera provides a deeper understanding of the data’s variability and the model’s performance. By managing and reducing entropy through various techniques, we can potentially improve the model’s ability to accurately classify frames, thereby enhancing the overall effectiveness of the punch detection system. This approach can boost the model’s reliability and ensure a more robust analysis suitable for real-world applications in sports analytics.

## 7. Conclusions

The analysis of the obtained results proved that punch detecting using one static RGB camera is possible. In order to achieve this, two techniques were combined: The first was to measure the distances between boxers in order to detect any clashes between them. The second one was more complex and was used to detect hit punches between boxers in clashes.

As the experiments showed, detecting punches using a classifier trained on unprocessed images yields subpar results. We propose three novel approaches to make recorded scenes easier for classification. The approach of removing static elements from the footage has the best scores in almost all calculated metrics.

We have also presented a working system for analyses of a boxing scene that marks the boxers, labels the frames with detected clashes and punches, and counts all situations with punches between the boxers. It is worth noting that the system is ready to automatically label recorded footage, for example, to auto-generate short clips from fights or for the need of broadcasting.

Despite these promising results, several significant shortcomings and open research directions were identified. One notable limitation was the reliance on a single static camera, which might not capture all relevant angles and details of the boxers’ movements. Future work could explore the integration of multiple camera channels to provide a more comprehensive view of the boxing ring, potentially enhancing the accuracy and robustness of the punch detection system.

Another limitation related to the classifier’s performance on unprocessed images underscores the need for more advanced image preprocessing techniques. Future research could investigate alternative preprocessing methods or the use of data augmentation to improve the classifier’s performance.

Future plans included synchronising data from all four cameras and preparing an ensemble voting framework to improve classification performance. There was also an aim to create a sophisticated neural network structure for punch classification. Further exploration into the use of temporal information, such as analyzing sequences of frames rather than individual frames, could enhance the system’s ability to detect punches more accurately.

In conclusion, while this study demonstrated the feasibility of using a single static camera for punch detection in boxing, numerous avenues for future research remain. Addressing these limitations and following these directions could lead to the development of more robust and accurate systems for sports analytics, ultimately benefiting athletes, coaches, and broadcasters.

## Figures and Tables

**Figure 1 entropy-26-00617-f001:**
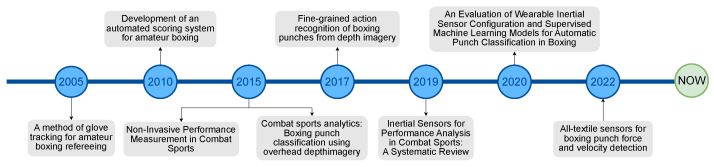
Timeline chart for the most relevant works to the study.

**Figure 2 entropy-26-00617-f002:**
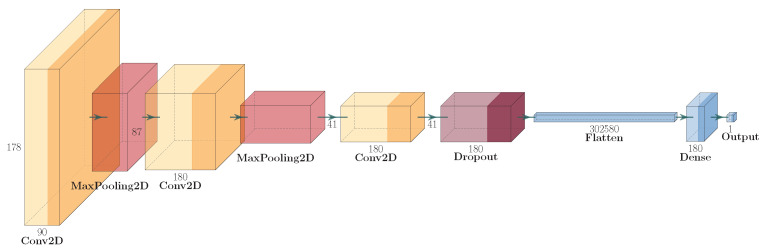
Structure of convolutional neural network used.

**Figure 3 entropy-26-00617-f003:**
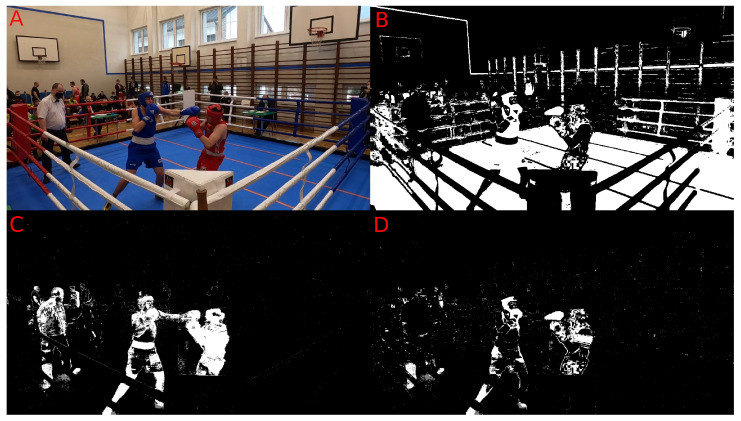
Visualisation of the original image along with the proposed approach of image manipulation. (**A**)—Original frame from the camera. (**B**)—Binary mask after segmentation based on colour extraction approach. (**C**)—Binary mask after segmentation based on background subtraction approach. (**D**)—Binary mask after segmentation based on hybrid approach.

**Figure 4 entropy-26-00617-f004:**
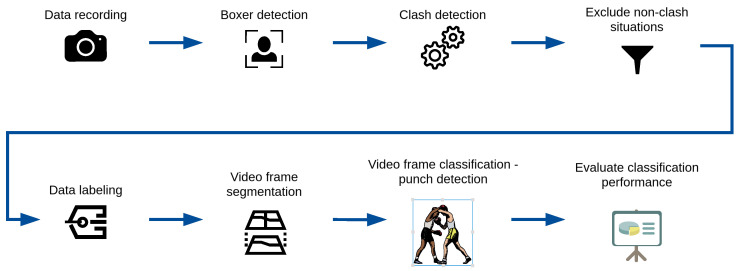
Methodology flowchart of the entire processing pipeline proposed in this study.

**Figure 5 entropy-26-00617-f005:**
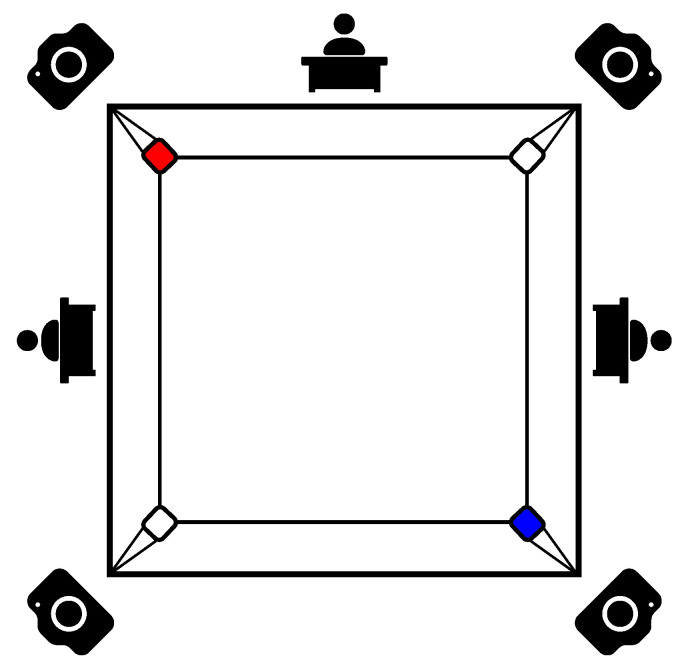
Diagram of the location of cameras and referees around the boxing ring.

**Figure 6 entropy-26-00617-f006:**
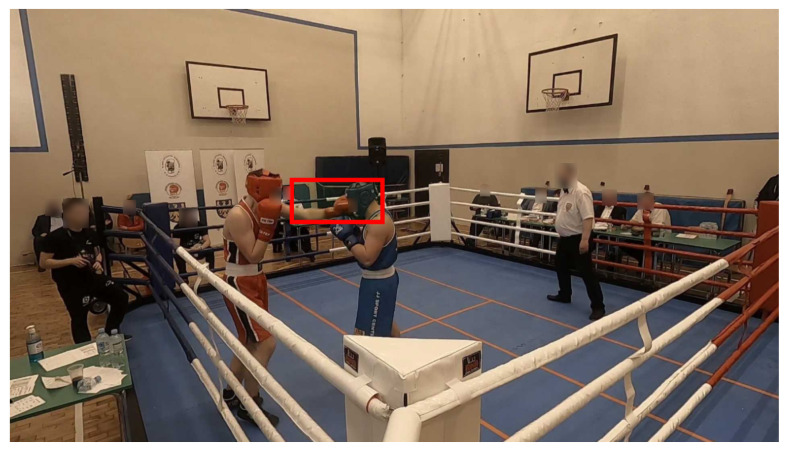
Example of a frame labelled as a punch by the referee.

**Figure 7 entropy-26-00617-f007:**
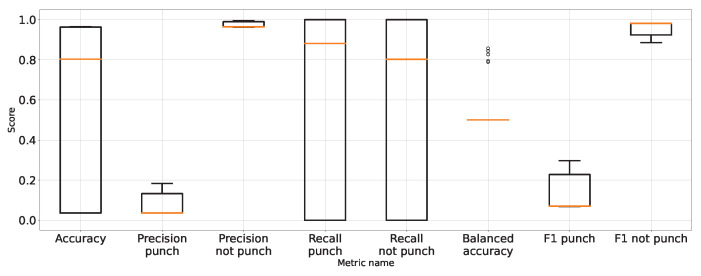
Performance of classification on original images.

**Figure 8 entropy-26-00617-f008:**
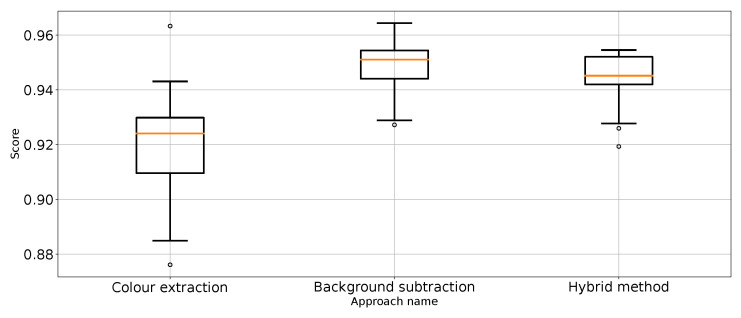
Accuracy for the three proposed approaches (one outlier was removed).

**Figure 9 entropy-26-00617-f009:**
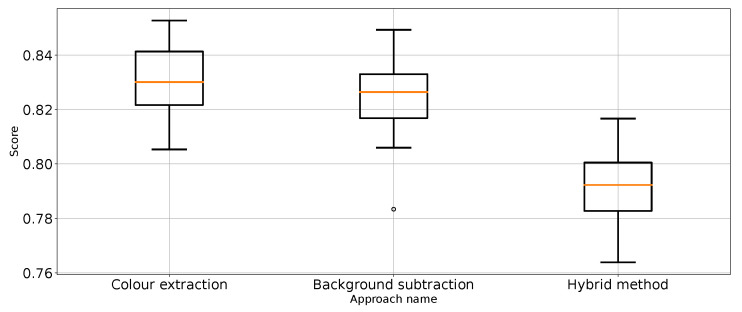
Balanced accuracy the for three proposed approaches (four outliers were removed).

**Figure 10 entropy-26-00617-f010:**
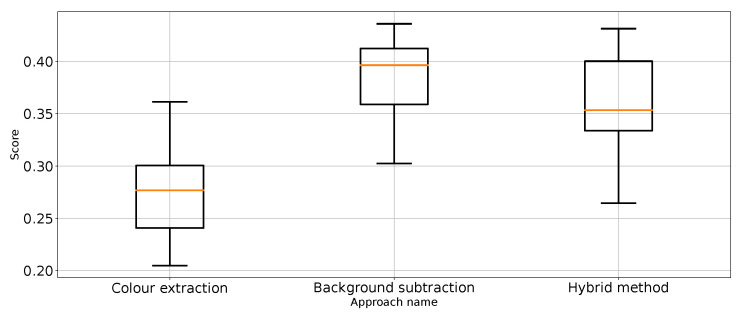
Precision for “punch” class for the three proposed approaches (one outlier was removed).

**Figure 11 entropy-26-00617-f011:**
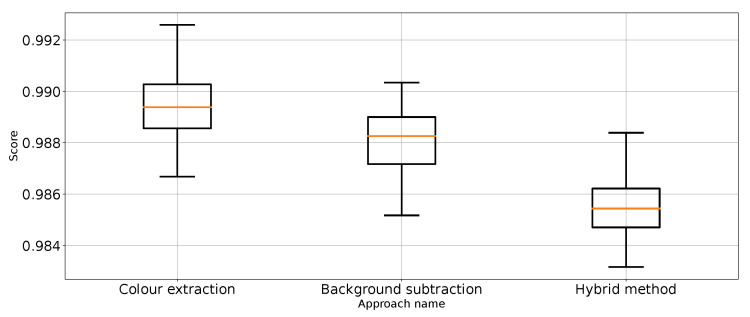
Precision for “no punch” class for the three proposed approaches (three outliers were removed).

**Figure 12 entropy-26-00617-f012:**
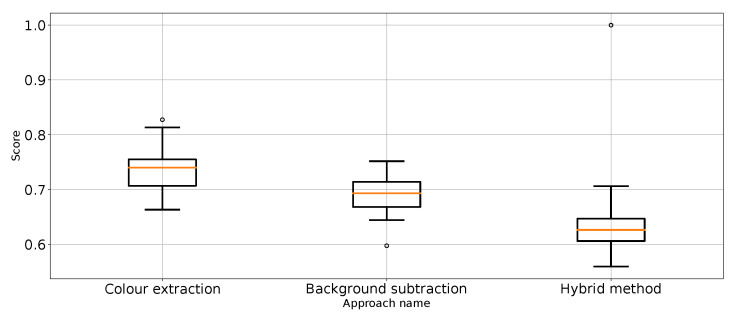
Recall for “punch” class for the three proposed approaches (three outliers were removed).

**Figure 13 entropy-26-00617-f013:**
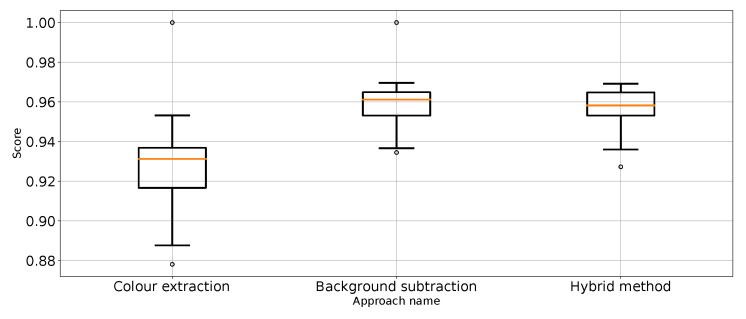
Recall for “no punch” class for the three proposed approaches (one outlier was removed).

**Figure 14 entropy-26-00617-f014:**
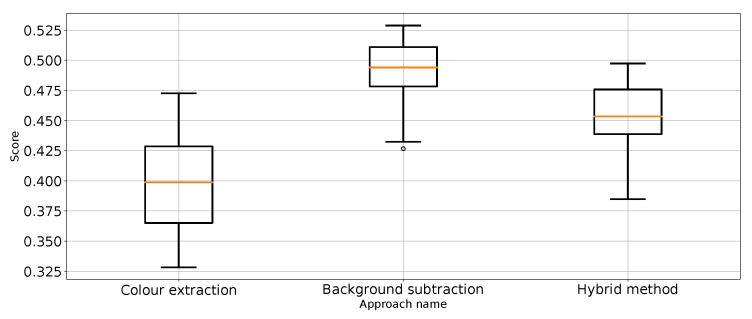
F1 value for “punch” class for the three proposed approaches (one outlier was removed).

**Figure 15 entropy-26-00617-f015:**
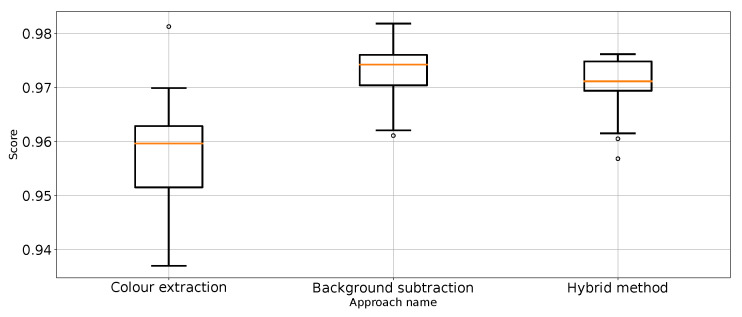
F1 value for “no punch” class for the three approaches (one outlier was removed).

**Figure 16 entropy-26-00617-f016:**
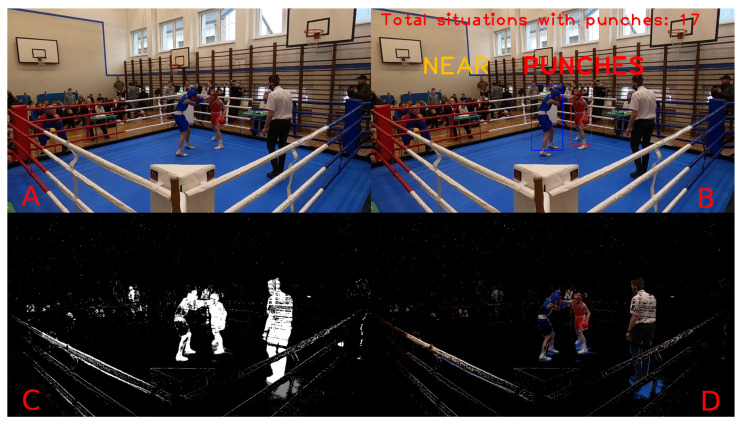
An overview of the whole processing pipeline for a punch event. (**A**): Original image; (**B**): original image with detected information; (**C**): mask of proposed Approach 3; (**D**): original image with detection mask applied.

**Figure 17 entropy-26-00617-f017:**
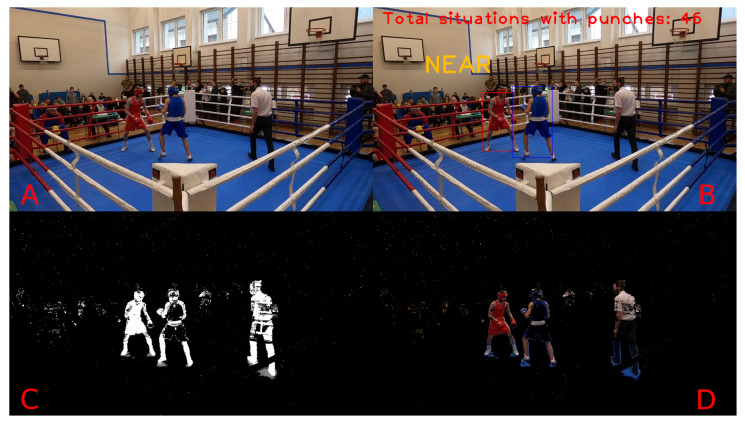
An overview of the whole processing pipeline for the close-combat situation without punches. (**A**): Original image; (**B**): original image with detected information; (**C**): mask of proposed Approach 3; (**D**): original image with detection mask applied.

**Figure 18 entropy-26-00617-f018:**
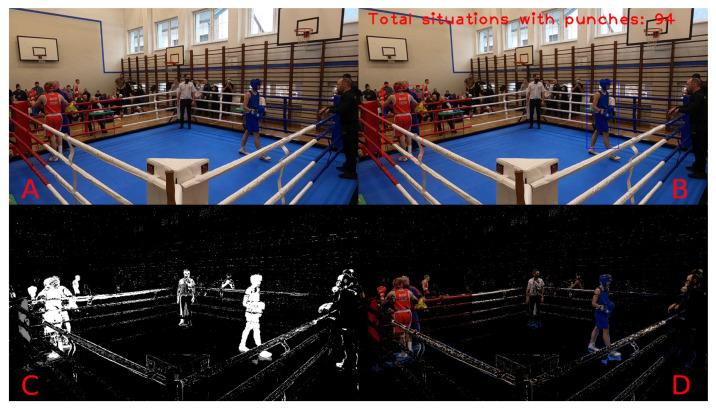
An overview of the whole processing pipeline for the no-contact situation. (**A**): Original image; (**B**): original image with detected information; (**C**): mask of proposed Approach 3; (**D**): original image with detection mask applied.

**Table 1 entropy-26-00617-t001:** Medians of classification performance metrics for all approaches.

Metric	Apr. 1	Apr. 2	Apr. 3	Apr. 4
Accuracy	0.8033	0.9240	0.9502	0.9449
Balanced accuracy	0.5000	0.8299	0.8257	0.7922
Precision punch	0.0364	0.2767	0.3899	0.3529
Precision not punch	0.9643	0.9893	0.9882	0.9854
Recall punch	0.8808	0.7377	0.6935	0.6260
Recall not punch	0.8023	0.9311	0.9600	0.9578
F1 punch	0.0703	0.3987	0.4932	0.4495
F1 no punch	0.9812	0.9594	0.9738	0.9711

**Table 2 entropy-26-00617-t002:** Medians of confusion matrix for all approaches.

Metric	Apr. 1	Apr. 2	Apr. 3	Apr. 4
TN	1965.500	1664.000	1560.500	1428.000
FP	266.000	597.000	690.500	848.500
FN	11,922.500	4151.500	2345.000	2539.500
TP	48,387.000	56,154.000	57,969.000	57,721.500

**Table 3 entropy-26-00617-t003:** Average processing time and resource utilisation for the tested approaches.

Metric	Apr. 2	Apr. 3	Apr. 4
Time (s)	161.115	272.234	356.519
CPU avg (%)	37.438	82.387	66.852
Mem avg (%)	72.593	63.760	70.404

**Table 4 entropy-26-00617-t004:** The Friedman test results and mean ranks.

	Values
**N**	8
**Chi-Square**	6.4500
**Degrees of Freedom**	3
***p*** **Value is Less Than**	0.0917
**5% Critical Difference**	1.2274
Mean Ranks	
**Approach 1**	3.2500
**Approach 2**	2.5000
**Approach 3**	1.6250
**Approach 4**	2.6250

## Data Availability

The original data presented in the study are openly available on Kaggle at https://www.kaggle.com/datasets/piotrstefaskiue/olympic-boxing-punch-classification-video-dataset (accessed on 15 May 2024).

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
