# Peer review of "Boxing Punch Detection with Single Static Camera"

_entropy, 2024, doi:10.3390/e26080617_

Round 1

Reviewer 1 Report

Comments and Suggestions for Authors

In this paper, the authors tackled an interesting problem of detecting boxing punches based on the image data acquired using a single camera. The topic is indeed interesting and worthy of investigation, and the presented ideas are valid. There are, however, quite a number of shortcomings which need to be thoroughly addressed before, in my opinion, the manuscript could be considered for publication:

1.       Although the manuscript reads well in general, it would still benefit from careful proofreading – I spotted quite a number of grammar errors around the manuscript, the authors should avoid using short forms (such as “doesn’t” – such forms are used even in the abstract), and there are typos arounds the manuscript (e.g., neurones”). Finally, the authors mix American and British English – it would be useful to keep the language concise. Also, please revise the tenses (I suggest to avoid using continuous tenses).

2.       There are unresolved references, e.g., to figures – see e.g., Figure XXX on page 1, line 35.

3.       While discussing the structure of the manuscript, it looks like the authors have missed Section 7. Also, the number of sections is rather large – perhaps it might be useful to consider restructuring the paper a bit.

4.       All entities in the manuscript (such as figures and tables) should convey useful information to the reader. As an example, Figure 1 may be easily removed from the manuscript, as it is rather “obvious”. Also, the authors should revisit all captions of such entities, and should expand them, in order to the these entities self-contained and possible to understand without diving into the text. As an example, it is unclear what are A-D in Figure 2.

5.       I would consider substantially squeezing the Background section and merging it with the Related works (or even dropping it entirely, as it contains rather basic and obvious information).

6.       I encourage the authors to slightly rework the related literature part of the manuscript, to clearly present the open research gaps in the field of using machine learning and computer vision in sports, with a special emphasis on boxing. Here, the authors should present, in a concise (tabular?) form a list of techniques available in the literature, showing their advantages and disadvantages. This would indeed help better contextualize the work within the state of the art, and should be helpful to better highlight the most important contributions behind the work reported here.

7.       It would be useful to present a high-level flowchart of the entire processing pipeline.

8.       I encourage the authors to present a figure showing the architecture of the CNN, rather than its textual description (page 4).

9.       What are the rules of the outfit colors regulated by the Olympic boxing?

10.  Please avoid having paragraphs containing  a single sentence only (see e.g., lines 227-8).

11.  Please present the annotation process in a visual flowchart, to indicate its most important steps. This would help a reader understand how the data was acquired and labeled. Also, I encourage adding some visual examples here.

12.  The training-test dataset splits remain unclear – there were 30 random splits following Monte-Carlo cross-validation, but how many (in %) images were included in both training and test sets?

13.  All figures in the manuscript should be high-quality in a vector format.

14.  We are currently facing the reproducibility crisis in the machine learning field. To address it, it would be fantastic if the authors could make their dataset (or at least a subset of it) publicly available, together with the scripts showing how to reproduce the numbers reported in this manuscript.

15.  It would be useful to analyze the non-functional capabilities of the proposed methods in more detail, with a special emphasis on inference time.

16.  What are the most important shortcomings and open research directions which might be followed based on the results of this research? It would be useful to discuss it in detail.

Comments on the Quality of English Language

See my main comments to the authors.

Author Response

All responses for remarks are attached as a file.

Reviewer 2 Report

Comments and Suggestions for Authors

This study proposes a method for detecting boxing punches using a monocular camera. The authors designed a CNN to identify punch actions and estimate the distance between two boxers. They also use various image processing techniques to alter the format of input images, in order to test the network's feasibility. The following are some format suggestions:

  1. Line 35: A figure reference is missing.
  2. Section 2 could be combined with Section 1. In addition, most of the information presented is well-known (they can be found in many textbooks).
  3. Section 3 could be enhanced. While the authors reviewed many computer vision literatures, some are not particularly relevant to this study. In addition, the motivation for this study should be clarified. The reasons for needing such a system, the current approach to the problem the authors are addressing, and the potential advantages of the proposed approach should all be discussed.
  4. Line 181: The input dimension about the proposed network is 180-by-180. However, the resolution of the input video frames are recorded in full HD. It is not clear why the authors resize the frame into such small size. By the way, it is also not clear why the authors choose a classification model instead of a detection model, as intuitively this problem should be a detection problem.
  5. In Figure 2, please describe subfigures A, B, C, D in the figure caption.
  6. For equation 2-5, these metrics are usually placed in the experiments section.
  7. The authors claim to use a single static camera to analyze boxers' action, but four cameras are placed around the ring as shown in Figure 3. The authors explained this is to prevent occlusions, yet in practice, it's challenging to know the exact time to switch to the correct camera.
  8. Figure 4 is mentioned in section 5, but it's placed in section 4.
  9. I suggest that the authors use confusion matrices to compare the performance of different approaches.
  10. The method of obtaining the detection mask in Figure 13 is not clearly stated.
  11. The authors mentioned imbalanced dataset in the manuscript. However, almost all detection problems are imbalanced and many state-of-the-arts are already proposed.

To sum up, The approaches used in this study are quite well-known and their efficacy has been proven by many previous works. Hence the contribution of this study is not obvious.

Comments on the Quality of English Language

This article is readable but the structure can be improved.

Author Response

(The authors gave the same response as above.)

Round 2

Reviewer 1 Report

Comments and Suggestions for Authors

Thank you for addressing my concerns.

Author Response

Thank you for your positive feedback on our manuscript. We are grateful for your comments and suggestions that helped to improve the quality and clarity of our work. Your feedback has been very helpful in developing the current version of our manuscript. Thank you again for your constructive comments and your time.

Reviewer 2 Report

Comments and Suggestions for Authors

As mentioned in the previous review comments, the major concern with this study is the lack of comparisons to the SOTA. Since the dataset is not public, we can only conclude that the proposed method may be effective, but we cannot say it is novel in terms of detecting a specific behavior using DNN. For example, a popular detection model may easily detect a boxer with a punch motion without as many steps as the proposed method in this study.

Author Response

(The authors gave the same response as above.)
